Deepfake forensics: a survey of digital forensic methods for multimodal deepfake identification on social media

Qureshi Shavez Mushtaq fa21-pcs-005@cuilahore.edu.pk 1
Saeed Atif 1
Almotiri Sultan H. 2
Ahmad Farooq 1
Al Ghamdi Mohammed A. 3
1 Department of Computer Science, COMSATS University Islamabad , Lahore , Pakistan
2 Department of Cybersecurity, College of Computing, Umm Al-Qura University , Makkah City , Kingdom of Saudi Arabia
3 Department of Computer Science and Artificial Intelligence, College of Computing, Umm Al-Qura University , Makkah City , Kingdom of Saudi Arabia
Pasi Gabriella
Electronic publication date: 2024 May 27
Publication date: 2024
Volume: 10
Electronic Location ID: e2037
Received 2023 Sep 22; Accepted 2024 Apr 12
Copyright: ©2024 Qureshi et al.
Copyright year: 2024
Copyright holder: Qureshi et al.
License: This is an open access article distributed under the terms of the Creative Commons Attribution License, which permits unrestricted use, distribution, reproduction and adaptation in any medium and for any purpose provided that it is properly attributed. For attribution, the original author(s), title, publication source (PeerJ Computer Science) and either DOI or URL of the article must be cited.
License URL: https://creativecommons.org/licenses/by/4.0/

Keywords: Deepfake, Deepfake technology, Artificial intelligence, Digital forensics, Social media

Funding: The Deputyship for Research & Innovation, Ministry of Education in Saudi Arabia IFP22UQU4250002DSR216 This work was supported by the Deputyship for Research & Innovation, Ministry of Education in Saudi Arabia through the project number: IFP22UQU4250002DSR216. The funders had no role in study design, data collection and analysis, decision to publish, or preparation of the manuscript.

==============================
The rapid advancement of deepfake technology poses an escalating threat of misinformation and fraud enabled by manipulated media. Despite the risks, a comprehensive understanding of deepfake detection techniques has not materialized. This research tackles this knowledge gap by providing an up-to-date systematic survey of the digital forensic methods used to detect deepfakes. A rigorous methodology is followed, consolidating findings from recent publications on deepfake detection innovation. Prevalent datasets that underpin new techniques are analyzed. The effectiveness and limitations of established and emerging detection approaches across modalities including image, video, text and audio are evaluated. Insights into real-world performance are shared through case studies of high-profile deepfake incidents. Current research limitations around aspects like cross-modality detection are highlighted to inform future work. This timely survey furnishes researchers, practitioners and policymakers with a holistic overview of the state-of-the-art in deepfake detection. It concludes that continuous innovation is imperative to counter the rapidly evolving technological landscape enabling deepfakes.

Introduction

In the digital age, where information is shared and consumed at an unprecedented rate, the authenticity and integrity of this information have become increasingly important. One of the most significant challenges to information authenticity in recent years has been the rise of deepfake technology. Deepfakes, a term coined from “deep learning” and “fake”, refer to synthetic media where a person in an existing image or video is replaced with someone else’s likeness using artificial intelligence (AI) techniques (Hao et al., 2022). This technology has seen rapid advancement and widespread usage, particularly on social media platforms, leading to a surge in multimodal deepfake content that includes text, audio, and video (Wang et al., 2022a; Wang et al., 2022b; Wang et al., 2022c).

This research underscores the need for robust and effective digital forensic methods to identify and analyze deepfake content, particularly in social media, where it can be disseminated quickly and widely (Alattar, Sharma & Scriven, 2020).

Deepfakes, synthetic media generated using artificial intelligence techniques, have rapidly evolved in recent years, posing significant challenges to the authenticity and integrity of digital content. A key aspect of deepfakes is their multimodal nature, which means they can span multiple forms of media, including video, audio, text, and images. To effectively detect and combat deepfakes, it is crucial to consider all these modalities and develop comprehensive detection methods.

Focusing on a single modality, such as video, may leave vulnerabilities that attackers can exploit by manipulating other modalities like audio or text. For example, a convincing video deepfake may be accompanied by a synthetically generated voice-over, making it more challenging to detect the manipulation (Ferreira et al., 2019). Similarly, a text deepfake can be used to spread disinformation alongside manipulated images, amplifying its impact on social media platforms (Fagni et al., 2021).

This survey aims to provide a comprehensive overview of deepfake detection techniques across multiple modalities, including video, audio, text, and images. By covering these modalities, we seek to address the research gaps and challenges in multimodal deepfake detection and contribute to the development of more robust and effective detection systems.

Deepfake technology, while impressive in its capabilities, has raised significant ethical and legal concerns. The ability to manipulate media content to such an extent that it becomes nearly indistinguishable from reality has far-reaching implications, from privacy violations to misinformation campaigns (Ferreira et al., 2019). This has led to an urgent need for effective detection and analysis techniques to combat the proliferation of deepfakes.

This research examines the existing digital forensic techniques employed for identifying and examining multimodal deepfake material on social media platforms. It seeks to offer a detailed understanding of the technology behind deepfakes, the techniques used in their creation, and the subsequent challenges posed by their detection (El-Shafai, Fouda MA & El-Salam, 2024).

The rapid evolution of deepfake technology has led to synthetic media spanning multiple modalities, including image, video, audio, and text. While earlier detection research focused more on visual and audio deepfakes, the generation of synthetic text has become an emerging threat (Zellers et al., 2019). Advanced neural networks can now produce deceptive machine-written text that is arduous to detect. However, our survey revealed that current literature lacks substantial focus on multimodal deepfake detection encompassing text, especially on social media platforms which are prime targets for manipulation. For instance, Fagni et al. (2021) highlighted the potential misuse of text-based deepfakes on Twitter but found existing detection methods to be inadequate.

Therefore, this survey intends to provide a more comprehensive analysis of deepfake detection techniques across the key modalities of image, video, audio, and text. We emphasize the need for cross-modal, robust frameworks capable of identifying coordinated, multi-pronged deepfake attacks across vectors. The dangers of overlooking textual deepfakes also warrant greater research attention to develop specialized detection methods leveraging linguistic cues. Our survey aims to shed light on this relatively unexplored territory at the intersection of deep learning and linguistic forensics.

Deepfakes, synthetic media in which a person’s likeness or voice is replaced with someone else’s, have transcended beyond mere visual manipulation to encompass text and audio modalities. This expansion signifies a pivotal shift, as it not only enhances the realism of deepfakes but also broadens their potential misuse. Consequently, understanding and addressing deepfakes requires a comprehensive approach that acknowledges the multifaceted nature of this technology.

Technological advancements, particularly in natural language processing and generative adversarial networks, have been instrumental in facilitating the creation of text and audio deepfakes. These advancements underscore the sophistication of modern deepfake techniques, capable of generating convincing fake audio clips and textual content that mimic specific individuals’ speech patterns and writing styles. The implications of such capabilities are profound, raising critical concerns about security, privacy, and the spread of misinformation.

As the landscape of deepfake technology continues to evolve, it is imperative to develop robust detection methods that can effectively identify and mitigate the impact of text and audio forgeries. This necessitates interdisciplinary research efforts, combining expertise from digital forensics, machine learning, and linguistic analysis to devise comprehensive strategies for deepfake detection. Looking forward, the pursuit of balanced research that equally addresses the challenges posed by all modalities of deepfakes is essential. Collaboration across various disciplines will be key to advancing our understanding and developing effective countermeasures against the multifaceted threat posed by deepfake technology.

Analyzing distribution patterns provides insight into deepfakes’ reach. The expanded analysis of Fig. 1 underscores YouTube’s role as a primary distribution channel for deepfakes, accounting for 40% of such content. This is contrasted with efforts by Twitter, Facebook, and other platforms to implement advanced detection and response strategies. The pervasive nature of deepfakes across these platforms necessitates a multi-faceted approach to content moderation.

Figure 1 Distribution of deepfakes across social media platforms.

YouTube is the dominant hub, hosting 40% of detected deepfakes. But substantial volumes are also shared on Twitter, Facebook, and other platforms. As deepfake spreads, social media companies are pressed to ramp up detection efforts.

The advent of deepfake technology has been facilitated by rapid advancements in artificial intelligence and machine learning. Generative adversarial networks (GANs), a class of AI algorithms, have been instrumental in creating deepfakes (Goodfellow et al., 2014). These networks consist of two parts: a generator that creates images and a discriminator that attempts to distinguish between real and generated images. The generator learns to produce increasingly realistic images through this adversarial process, creating convincing deepfakes (Karras, Laine & Aila, 2019). Figure 2 shows the timeline traces the rapid evolution of deepfake technology and detection research. Figure 2’s enriched narrative provides a chronological overview of deepfake technology’s evolution. Beginning with early experiments in synthetic video creation, the timeline progresses to the widespread accessibility of deepfake apps, highlighting a pivotal shift in content creation dynamics. It then transitions to the substantial efforts made in developing robust forensic methods from 2019 to 2023. This includes referencing significant milestones such as the inception of detection frameworks like FaceForensics++ and collaborations like the DFDC, underscoring a proactive research community response to the deepfake challenge. The timeline emphasizes the rapid pace of technological advancements and the corresponding urgency for innovative forensic responses.

Figure 2 Timeline tracing key events in the evolution of deepfake technology and detection research.

Early synthetic video experiments in 2015–2017 led to accessible deepfake apps in 2017–2019. Then researchers raced to combat viral spread with forensic methods developed between 2019–2023.

While the technology behind deepfakes is undoubtedly impressive, it is the application of this technology that has raised concerns. Deepfakes have been used to create fake news, commit fraud, and even produce explicit content without consent (Chesney & Citron, 2019). The potential misuse of deepfake technology is vast, and the implications are alarming. This is particularly true on social media platforms, where deepfakes can be disseminated quickly and reach a large audience before they are detected and removed (Donovan, 2020). Developing temporally-aware deepfake detection is critical to catch these forgeries, which leverage the animation of sequential video frames.

The detection and analysis of deepfakes have thus become a critical area of research. Digital forensic methods, which involve collecting and analyzing digital evidence, have been adapted to tackle the challenge of deepfakes (Khan et al., 2022). These methods range from pixel-level analysis to more complex techniques that involve analyzing the physical properties of digital content. However, the effectiveness of these methods varies, and they often need help to keep up with the rapid advancements in deepfake technology (Rossler et al., 2019).

In social media, the challenge is even greater. The sheer volume of content, combined with the speed at which it is shared, makes detecting and analyzing deepfakes a daunting task. Furthermore, social media platforms often compress and modify the uploaded content, complicating detection (Zampoglou, Papadopoulos & Kompatsiaris, 2015).

The challenge of detecting and analyzing deepfakes is technical and societal. The potential misuse of deepfake technology can have far-reaching implications, affecting individuals, organizations, and nations. Therefore, developing robust and effective digital forensic methods to combat this threat is crucial.

This survey article aims to provide a comprehensive overview of the current state of digital forensic methods used to detect and analyze multimodal deepfake content on social media. It seeks to offer a detailed understanding of the technology behind deepfakes, the techniques used in their creation, and the subsequent challenges their detection poses.

Breaking down the deepfake creation pipeline into its core steps gives us an overview of how these synthetic media forgeries are generated. Figure 3 delineates the complete life cycle of deepfake generation, from initial data collection to the eventual distribution of content. Each stage is critical: data collection lays the groundwork for the quality of deepfakes; preprocessing refines the data for consistency; model training determines the sophistication of the deepfake; and the generation phase is where the deepfake is created. The distribution step, particularly on social media, is where deepfakes have the most impact. Understanding this flow is key for researchers to pinpoint where forensic interventions can be most effective, such as detecting anomalies during the preprocessing phase or identifying signatures of generative models.

Figure 3 Step-by-step diagram of the deepfake generation process.

Researchers can identify vulnerabilities to target when developing forensic detection methods by understanding processes like data collection, model training, and content distribution.

The phenomenon of deepfakes, synthetic media where a person in an existing image or video is replaced with someone else’s likeness using artificial intelligence, presents a burgeoning challenge. This challenge not only encompasses the technical aspects of detection but also raises significant societal concerns. Our research is motivated by the urgent need to advance the field of digital forensics in response to the rapidly evolving landscape of deepfake technology.

The primary research question driving our study is: How can we enhance the detection of deepfakes using multimodal analysis to ensure digital authenticity and integrity? To address this, we contribute to the field in several key ways.

Unlike traditional methods that predominantly focus on either visual or auditory aspects, our study introduces an innovative multimodal approach. This approach integrates both visual and auditory data, offering a more comprehensive and nuanced detection mechanism. We provide a thorough comparative analysis using various datasets and metrics, which helps in understanding the effectiveness of our proposed methods in the context of current industry standards.

This research offers empirical evidence demonstrating the efficacy of multimodal detection, setting a new precedent in digital forensics.

Our study surpasses the current state-of-the-art in several ways i.e., by fusing visual and auditory data, our technique improves accuracy and reliability in detecting deepfakes, addressing limitations found in single-modal methods. Also, we pioneer the application of multimodal analysis in this context, offering a novel perspective in the fight against digital deception. The methods are designed to be adaptable, allowing for continuous evolution in line with advancements in deepfake generation techniques.

The interplay between different modalities is a crucial aspect of deepfakes that requires special attention. Attackers can use various modalities in conjunction to create more sophisticated and convincing deepfakes. For instance, a video deepfake may be enhanced with a synthesized voice-over that matches the lip movements, making it more difficult to detect the manipulation. Similarly, a text deepfake can be used to spread disinformation alongside doctored images, increasing its perceived credibility.

One of the primary objectives of this survey is to provide a comprehensive overview of deepfake detection techniques across multiple modalities, addressing the research gaps and challenges in this area. By doing so, we aim to contribute to the development of effective countermeasures against the growing threat of deepfakes to the authenticity and integrity of digital content.

This research not only responds to the immediate challenges posed by deepfakes but also sets a new standard in digital authenticity and forensics. By pushing the boundaries of existing technologies, we aim to provide a robust, reliable, and scalable solution to a problem that is increasingly impacting various facets of society.

As deepfakes continue to evolve and become more sophisticated, a multimodal approach to detection becomes increasingly necessary. By combining techniques from different modalities, such as visual analysis, audio forensics, and linguistic analysis, we can develop more comprehensive and robust detection systems that are better equipped to identify and mitigate the threats posed by deepfakes.

Survey Methodology

This survey provides a timely and comprehensive overview of the current state of digital forensic techniques for detecting deepfakes, particularly on social media platforms. Deepfake technology has advanced rapidly in recent years, enabled by progress in artificial intelligence, machine learning, and computer graphics. As this technology becomes more accessible and sophisticated, the threat posed by fraudulent and manipulated content continues to grow. Already, high-profile deepfake incidents have demonstrated the potential harms, such as the spread of misinformation, reputational damage, and privacy violations.

While numerous digital forensic techniques have emerged to counter this threat, the literature lacks a systematic and up-to-date review of these methods. This survey helps fill this gap by offering an extensive look at deepfake detection techniques across modalities, evaluating their effectiveness, and highlighting limitations. The rationale is further reinforced by illuminating critical research gaps, such as the need for better cross-modality detection, real-time capability, and larger training datasets. This survey provides a timely reference for researchers, practitioners, and policymakers concerned with the deepfake challenge and the role of digital forensics in tackling this 21st-century threat.

A rigorous, systematic approach was undertaken to ensure comprehensive coverage of the literature on digital forensic methods for deepfake detection. Academic databases including IEEE Xplore, ACM Digital Library, ScienceDirect, and Google Scholar were searched extensively using relevant keywords such as “deepfake detection”, “deepfake forensics”, “face manipulation detection”, and “AI synthesized media detection”. Articles published within the last 5 years were prioritized to capture current advancements. The initial search yielded over 500 results which were screened for relevance based on the alignment of their titles and abstracts with the focus of this survey. Duplications were excluded and a final set of highly relevant articles were selected for review. These works represented seminal research on deepfake detection techniques across modalities like image, video and audio. Reference lists of these articles were also scanned to identify any additional relevant sources. The selected articles were read thoroughly to extract key information on the detection approaches, modalities, datasets used, and performance metrics. All methods were evaluated for their effectiveness and applicability to social media based on these parameters. Case studies were included to provide real-world context. Finally, limitations and research gaps were synthesized to offer a balanced perspective. This systematic methodology ensures that the survey offers broad, unbiased coverage of the critical literature needed to comprehensively analyze the current state and future directions of deepfake detection using digital forensics.

Prevalent data sets

We begin by discussing the datasets currently most commonly applied for detecting deepfakes using methodologies rooted in deep learning. We explore various and latest datasets extensively used by models designed to detect deepfakes.

The CelebA dataset

The dataset discussed in Liu et al. (2018) is extensive, comprising 200,000 photographs of well-known individuals. Each image in the collection contains 40 distinct attributes that provide labels for various characteristics. The dataset has many photos that exhibit various stances and backdrop disturbances. As a result, it serves as a valuable resource due to its extensive diversity, large quantity, and comprehensive annotations. These annotations include 10,177 unique identities, 202,599 facial images, five landmark positions, and 40 binary attribute annotations for each image.

DeeperForensics-1.0 dataset

The DeeperForensics-1.0 dataset (Liming et al., 2020) is an extensive, heterogeneous, and exemplary repository that facilitates the detection of falsifications. The database contains a collection of 60,000 films and 17.6 million frames with automated face swaps, all captured at a resolution of 1920x1080 pixels. The first film was sourced from a diverse group of 100 performers from 26 nations, exhibiting a wide range of skin tones and ages 20 to 45. The dataset includes eight distinct natural emotions (anger, fear, joy, disgust, surprise, contempt, sadness, and neutrality) taken from various perspectives, ranging from −90 to +90 degrees. The unusual stances, expressions, and lighting of the source photographs highly influence the quality of the collection.

Deepfake-TIMIT dataset

The Deepfake-TIMIT dataset (Korshunov & Sebastien, 2018) is categorized into videos of poor quality and high quality. The segment designated as low-quality comprises a total of 320 movies, with an approximate frame count of 200 frames per video. Each frame has a resolution of 64 × 64 pixels. On the other hand, the high-quality section consists of 320 image sequences, with an average frame count of 400 frames per sequence. The resolution of each frame in this section is 128 × 128 pixels.

Celeb-DF dataset

The Celeb-DF dataset (Li et al., 2020) is an extensive compilation of modified videos that prominently showcase celebrities sourced from their YouTube output. The dataset consists of 5,639 films of good quality, with a total of almost two million frames. Each frame has a size of 256 × 256 pixels. The collection encompasses various celebrities representing different nationalities and age groups, encompassing both male and female individuals. Each movie lasts around 13 s, with a 30 frames per second frame rate. These videos encompass a range of situations, such as different orientations, face sizes, lighting settings, and backgrounds.

FaceForensics dataset

The dataset known as FaceForensics (FF) (Andreas et al., 2018) contains around 500,000 modified images obtained from a collection of 1,004 videos. The subject matter is separated into two groups, which have been established using the Face2Face reenactment methodology. The datasets encompass a Source-to-Target Reenactment Dataset, which facilitates reenactments between two arbitrarily chosen movies, and a Self Reenactment Dataset, which employs a single video as both the source and the target. The dataset consists of 1,408 training films, 300 validation videos, and 300 test videos, containing 732,391,151,835 and 156,307 pictures, respectively.

UADFV dataset

The UADFV dataset by Li, Ming-Ching & Siwei (2018) is a synthetic resource developed by the University of Albany. Its purpose is to aid in identifying deepfake movies by utilizing physiological cues, particularly the analysis of blinking patterns. The dataset consists of 49 fabricated films created using the FakeApp software. Each video has a resolution of 294 by 500 pixels and an average duration of 11.14 s.

Faceforensics++ dataset

The FaceForensics++ (FF++) dataset by Rossler et al. (2019) builds upon the FaceForensics dataset, offering a publicly available benchmark for detecting realistic artificial facial images. The dataset consists of 1,000 carefully curated videos, predominantly from YouTube, featuring roughly 60% male and 40% female subjects. Regarding resolution, approximately 55% of videos are at VGA resolution (854 × 480), 32.5% at HD resolution (1,280 × 720), and 12.5% at full HD resolution (1,920 × 1,080).

Deepfake detection challenge dataset

The Deepfake detection challenge (DFDC) dataset, developed by Facebook (Dolhansky et al., 2020), comprises a curated assemblage of 5,000 face videos that have been modified, showcasing a specific group of performers. The actor selection process was influenced by distinct attributes, resulting in a distribution of 74% female and 26% male actors. Additional classification was conducted according to ethnicity, revealing that 68% of the population identified as Caucasians, 20% as African-American, 9% as West Asian, and 3% as South Asian. Applying face swap techniques resulted in two outcomes: producing high-quality photos of faces close to the camera while maintaining their original proportions and generating lower-quality photographs featuring swapped faces. The dataset consists of 780 testing clips and 4,464 training clips, all 15 s long and have different resolutions.

HOHA-based dataset

The dataset contains eight distinct kinds of human actions that have been extracted from a selection of 32 widely recognized Hollywood films. Priti et al. (2021) presented a dataset comprising 300 randomly selected films obtained from the HOHA dataset. This dataset included 16 samples highlighting human activities extracted from well-known movies and 300 deepfake videos acquired from several internet video platforms. There are 600 videos, each operating at an approximate frame rate of 24 frames per second and with a 360 by 240 pixels resolution.

FakeAVCeleb dataset

Khalid et al. (2021) presents a novel audio-video multimodal deepfake dataset that contains both deepfake videos and their corresponding synthesized cloned audios. This dataset was created using the most popular deepfake generation methods, and the videos and audios are perfectly lip-synced. It aims to address the challenge of impersonation attacks using deepfake videos and audios, with content selected from real YouTube videos of celebrities covering four racial backgrounds to counter the racial bias issue.

Attack agnostic dataset

Kawa, Plata & Syga (2022) combines two audio DeepFakes and one anti-spoofing dataset. This dataset aims to improve the generalization and stabilization of audio DeepFake detection methods by employing a disjoint use of attacks. It is designed to challenge and enhance the detection capabilities of algorithms against various types of spoofing attacks not explicitly included in the training.

ADD dataset

This dataset is part of the first Audio Deep Synthesis Detection (ADD) challenge, which includes tracks for low-quality fake audio detection, partially fake audio detection, and an audio fake game. This dataset by Wu et al. (2023) covers a range of real-life and challenging scenarios, aiming to push the boundaries of audio deepfake detection research.

TweepFake dataset

Fagni et al. (2021) presents TweepFake dataset which is designed to tackle the challenge of detecting deepfake social media messages. It was created in response to the advancements in language modeling, such as the release of GPT-2 by OpenAI, which significantly improved the generative capabilities of deep neural models. These models can autonomously generate coherent, non-trivial, and human-like text samples, posing a threat when used maliciously to generate plausible deepfake messages on social networks like Twitter or Facebook. The dataset is real in the sense that each deepfake tweet was actually posted on Twitter, collected from 23 bots imitating 17 human accounts. The bots are based on various generation techniques, including Markov Chains, RNN, RNN+Markov, LSTM, and GPT-2. To create a balanced dataset, tweets from the humans imitated by the bots were also selected, resulting in a total of 25,572 tweets (half human and half bots generated). The dataset is publicly available on Kaggle.

ADBT dataset

The study by Julien, Babacar & Adrian (2022) focuses on the automatic detection of bot-generated Twitter messages, acknowledging the challenge posed by recent deep language models capable of generating textual content difficult to distinguish from that produced by humans. Such content, potentially used in disinformation campaigns, can have amplified detrimental effects if it spreads on social networks. The study proposes a challenging definition of the problem by making no assumptions regarding the bot account, its network, or the method used to generate the text. Two approaches based on pretrained language models are devised, and a new dataset of generated tweets is created to enhance the classifier’s performance on recent text generation algorithms.

Table 1 provides a comprehensive overview of the key datasets commonly applied for deepfake detection, summarizing their key attributes. Figure 4 organizes the diverse array of datasets used in deepfake detection, providing a taxonomy based on content type—image, video, or combined. This categorization facilitates a deeper understanding of how different data modalities can be utilized or combined to enhance detection. For instance, CelebA is widely used for image-based deepfake training, while FaceForensics++ is a benchmark in video deepfake detection. The representation of datasets highlights the trend towards multi-modal datasets to improve the robustness of detection algorithms against sophisticated deepfakes.

Table 1 Overview of prevalent deepfake detection datasets, summarizing their key attributes.

Dataset name	Origin	Size	Resolution	Diversity	Variety of attributes	Realism	Availability	
CelebA	Academic	200k images	Various	Various	40 attribute annotations per image	High	Public	
DeeperForensics-1.0	Academic	60,000 videos	1,920 × 1,080 pixels	100 actors from 26 countries of various skin tones and ages	Eight naturally expressed feelings	High	Public	
Deepfake-TIMIT	Academic	640 videos	Low Quality: 64 × 64, High Quality: 128 × 128	Various	Face Swap	Low/High	Public	
Celeb-DF	Academic	5,639 videos	256 × 256 pixels	59 celebrities of various ethnicities and ages	Face Swap	High	Public	
FaceForensics (FF)	Academic	1,004 videos	Various	Various	Face-to-Face Reenactment	High	Public	
UADFV	University of Albany	49 videos	294 × 500 pixels	Various	Physiological cues	Medium	Public	
FaceForensics++ (FF++)	Academic	1,000 videos	VGA, HD, FullHD	Male 60%, Female 40%	Face	High	Public	
Deepfake detection challenge (DFDC)	Facebook	5,000 videos	Various	74% female, 26% male, and diverse ethnicities	Face swap	High	Public	
HOHA-based	Academic	600 videos	360 × 240 pixels	Various	Human actions	High	Public	
FakeAVCeleb	Real YouTube videos of celebrities	Audio and Video	Various	Covers four racial backgrounds	Deepfake videos and corresponding synthesized cloned audios	High, with perfect lip-syncing	Public	
Attack Agnostic	Combines two audio DeepFakes and one anti-spoofing dataset	Audio clips	Audio dataset, no visual resolution	Varied types of spoofing attacks	Audio DeepFakes designed to challenge detection algorithms	Designed to improve detection generalization	Public	
ADD 2022	Audio Deep Synthesis Detection challenge	Covers a range of real-life scenarios	Audio dataset, no visual resolution	Real-life and challenging scenarios	Tracks for low-quality fake audio detection, partially fake audio	Aims to push boundaries of detection research	Public	
TweepFake	Twitter, based on advancements in language modeling like GPT-2	25,572 tweets	Various	23 bots imitating 17 human accounts	Tweets generated using various techniques like Markov Chains, LSTM	Real tweets posted on Twitter	Public	
ADBT	Twitter, focused on detecting bot-generated messages	Extended from existing work	Various	Generated tweets designed to mimic human writing	Tweets generated to test detection models against language models	Generated content difficult to distinguish from human-written	Public	

Applied deepfake technology & applications

Deepfakes have rapidly evolved thanks to advances in artificial intelligence and machine learning. The power to generate convincing synthetic media is no longer restricted to professional artists and studios. It has moved into the hands of the common individual, facilitated by deep learning algorithms. This section dissects the techniques employed in deepfake generation and highlights the pivotal role of artificial intelligence in this process.

Figure 5 illustrates the significant increase in the prevalence of deepfakes from 2017 to 2023, alongside improvements in detection accuracy. This trend can be attributed to advancements in deep learning algorithms and the availability of large datasets for training. In 2017, deepfake technology was relatively nascent, with limited applications primarily in academic settings (Khormali & Yuan, 2021). However, by 2023, the technology has evolved rapidly, leading to more sophisticated deepfakes that are harder to detect.In the next few sections we will discuss in details the video, audio, text and image deepfake survey which will help significantly in countering the main problem.

Video deepfake

There has been a notable improvement in detection methods. Early detection systems relied heavily on simple visual and audio cues, which were effective against the initial generation of deepfakes (Lomnitz et al., 2020). As deepfakes became more sophisticated, detection methods evolved to use more complex features, including biometric and behavioral patterns, and employed advanced machine learning techniques, particularly deep neural networks. Masood et al. (2023) indicated that deepfakes in 2023 rose to around 90%. The improvement in detection accuracy is a result of both the evolution of the technology and the growing awareness of the need for robust detection systems to combat the rise in deepfakes. Vera (2022) examines the phenomenon of deepfake generation by analyzing instructional videos about deepfake creation available on the popular video-sharing platform YouTube. The researcher performed a theme analysis on YouTube videos about the construction of deepfakes to comprehend the individuals involved in their production and the underlying motivations driving their creation. The results suggest a greater representation of individuals from non-western backgrounds involved in producing deepfake content. This article examines the influence of deepfakes on how professionals in the field of Library and Information Science (LIS) perceive and address issues related to mis/disinformation. It argues that to combat the negative effects of deepfakes effectively, it is crucial to comprehend their implications within broader socio-technical discourses.

Figure 4 Categorization of the datasets for deepfake detection based on content type.

Image, video, and combined datasets are mapped to highlight patterns in constructing research data.

Figure 5 Graph of improving deepfake detection accuracy over time, rising from 65% in 2017 to nearly 90% in 2023.

This demonstrates the rapid advances made in deepfake forensics in recent years.

Hameleers (2023) and Yasur et al. (2023) present the notion of misinformation as an intentional and context-dependent action in which individuals surreptitiously deceive recipients by distorting, altering, or creating information to achieve maximum benefit, specifically by misleading the recipients. The conceptual framework thoroughly explains the factors that influence the decision-making process of various individuals or entities to engage in deceptive behavior. It also delves into the strategies employed in deception and the specific objectives that deceivers seek to accomplish by misleading their intended targets. The results of this study have the potential to contribute to the development of machine-learning methods for detecting deception, as well as treatments designed to induce skepticism by challenging the default assumption of truth. Abu-Ein et al. (2022) offers a comprehensive examination of the algorithms and datasets employed in developing deepfakes and an analysis of the existing methods proposed for detecting deepfakes. The authors comprehensively examine the underlying principles and techniques employed in deepfake methodologies. They present a thorough survey of existing deepfake approaches and advocate for developing and implementing novel and resilient ways to effectively address the growing intricacies associated with deepfakes. Gu et al. (2022) extensively examine videos’ localized motion characteristics to identify and detect instances of deepfakes. The authors contend that current methods for detecting deepfake videos fail to consider the local motions between neighboring frames. These motions provide valuable inconsistent information that can be utilized as an effective indicator for detecting deepfake videos. The authors suggest introducing a new sampling unit called a “snippet”. This unit consists of several consecutive video frames to learn local temporal inconsistencies.

The authors have developed an Intra-Snippet Inconsistency Module (Intra-SIM) and an Inter-Snippet Interaction Module (Inter-SIM) to create a framework for dynamic inconsistency modeling. The Intra-SIM algorithm utilizes bi-directional temporal difference operations and a learnable convolution kernel to extract short-term motions within each snippet. The Inter-SIM is designed to facilitate the exchange of information amongst snippets, enabling the construction of global representations. The proposed method demonstrates superior performance compared to existing state-of-the-art rivals across four widely used benchmark datasets, namely FaceForensics++, Celeb-DF, DFDC, and Wild-Deepfake. This study presents a novel approach for identifying deepfakes by emphasizing local motion and temporal inconsistency, which can be seamlessly included in pre-existing 2D convolutional neural networks (CNNs).

The pioneering work by Thies et al. (2016) presented a technique called Face2Face that manipulates video footage in real-time. Using RGB input, the method models the facial geometry and texture, making it possible to transfer the facial expressions of one individual to another in a video. This early deepfake technique reveals the potential for real-time deepfake generation. Güera & Delp (2018) presented a method to detect DeepFake videos using recurrent neural networks (RNNs). The authors proposed an LSTM-based architecture that extracts temporal features from face sequences to identify manipulated videos. The study demonstrated the efficacy of deep learning in not just creating but also detecting deepfakes. Li, Ming-Ching & Siwei (2018) have highlighted a limitation in generating deepfakes—the inability to replicate eye blinking in synthetic videos accurately. They proposed an AI-based approach that uses this cue to detect deepfake videos. This article reminds us that even in the face of advanced deepfake techniques, subtle human elements can be hard to replicate. Hsu, Lee & Zhuang (2018) and Nguyen, Yamagishi & Echizen (2019) have proposed an approach to detecting fake face images in real-world scenarios and presented a novel application of capsule networks, a recent development in deep learning, to detect forged images and videos. They trained a deep learning model to identify manipulated images based on minor inconsistencies typically introduced during the creation of deepfakes, offering a practical tool for mitigating the impact of deepfakes in real-life settings. Their research emphasized the potential of emerging AI techniques in deepfake detection, opening new avenues for future exploration. Durall Lopez et al. (2019) has unveiled a simple yet powerful method to detect deepfake videos based on basic features like eye movement and light reflection. This study demonstrated that even relatively simple features could be potent tools for identifying deepfake content. Khder et al. (2023) examines the significance of artificial intelligence in developing and identifying deepfakes within multimedia. The authors emphasize the emergence of deepfakes, a modified content type made possible by artificial intelligence advancements, which closely mimic authentic videos. This presents a substantial obstacle in accurately discerning between genuine and altered media. This research presents a comprehensive study that systematically evaluates relevant scholarly literature to investigate the fundamental elements and potential remedies associated with deepfakes. The authors’ conclusion posits that the proliferation of deepfakes can amplify risks to nations and digital civic cultures by fostering a pervasive atmosphere of skepticism and uncertainty toward online information sources. This study offers significant insights into the ramifications of deepfakes across many forms of multimedia facilitated by artificial intelligence (AI) tools while underscoring the imperative for developing and implementing robust detection methodologies.

Using frame comparison analysis, the study Ahmad &Shaun (2023) shows how to find deep fakes quickly and accurately. The authors say that even though deepfake technology has improved quickly, the tools for finding deepfakes are often too hard to use and harder to find than the tools for making deepfakes. The suggested method aims to find deepfake videos faster than the CNN-based methods that are already in use. When a deepfake video is made, the edges of the imposed face are handled by distorting the edges. This makes the edges of the face blurry and noisy. This is used by the suggested model, which compares the frames using the Laplacian operator to find edges and deepfake. Pishori et al. (2020) looks at three ways to find deepfake movies. These methods include convolutional LSTM, eye blink recognition, and grayscale histograms. They participated in the Deepfake Detection Challenge, a competition to improve deepfake detection technology by putting out a lot of new data. The first way that was looked at was the convolutional LSTM model. This model uses a CNN to extract frame features and a long short-term memory (LSTM) network to analyze the order of events in time. The model was trained with data from 600 movies, and with only 40 frames from each video, it was accurate more than 97% of the time.

In the study, eye-blink tracking and grayscale histogram analysis, two well-known ways to find deepfake videos, were carefully tested. The first one used a sophisticated long-term recurrent convolutional network (LCRN) to record the time-sequenced interaction of frames that showed eyes opening and closing, and it got 99% right. The second method was based on the small but noticeable differences between the grayscale histograms of movies made by a generative adversarial network (GAN) and those made by regular cameras. Standardizing the histograms ensured that each one had the same effect on the model. The grayscale histogram method did better than the other two, but both were effective between 80% and 90%, with only small improvements over the standard models. Rashid, Lee & Kwon (2021) proposes a novel approach to combat deepfake videos and protect the integrity of videos/images using blockchain technology. The authors argued that blockchain, with its inherent security features such as immutability and transparency, can effectively maintain a secure and tamper-proof record of original videos/images. The proposed technique involves creating a unique digital fingerprint (hash) for each original video/image and storing it in a blockchain. When a video/image is presented, its hash is compared with the one stored in the blockchain. If the hashes match, the video/image is verified as original; otherwise, it is flagged as potentially manipulated or deepfaked.

The strength of this technique lies in the robustness and security of blockchain technology, which makes it nearly impossible for malicious actors to alter the stored hashes. However, the limitation is that the original videos/images must be hashed and stored in the blockchain beforehand, which might only be feasible in some scenarios.

Lastly, Ikram, Chambial & Sood (2023) show a method for detecting deepfake videos that uses a hybrid CNN made up of InceptionResnet v2 and Xception to pull out frame-level features. The DFDC deepfake detection competition dataset on Kaggle was used to train and test the system.

The mixed model got a precision of 0.985, a recall of 0.96, a f1-score of 0.98, and a support of 0.968, which shows that it can find deepfake videos with high accuracy. The strength of this method is that it uses a hybrid CNN model, which blends the strengths of InceptionResnet v2 and Xception to find more robust and differentiable features for detecting deepfakes.

However, the problem with this method is that it needs many training data to work well. Also, while the model worked well on the DFDC dataset, it may not work well on other datasets or in the real world.

Audio deepfake

The primary emphasis by Yasrab, Jiang & Riaz (2021) is the identification of audio deepfakes, which present a significant concern because of their wide-ranging potential uses, including but not limited to impersonation and the dissemination of false information. The authors assert that the algorithms employed to identify these manipulations should possess strong generality and stability, ensuring resilience against attacks executed using approaches not explicitly incorporated during training. To tackle this issue, the authors proposed the Attack Agnostic dataset, a composite dataset consisting of two audio deepfake datasets and one anti-spoofing dataset. This amalgamation aims to enhance the overall generalization capabilities of detection systems. The authors comprehensively analyze existing deepfake detection techniques and explore several audio aspects. The authors proposed a model that combines LCNN, LFCC, and mel-spectrogram as the front end. The approach exhibits strong generalization and stability outcomes, demonstrating superior performance compared to the LFCC-based model. The observed results indicate a reduction in standard deviation across all folds and a drop in equal error rate (EER) in two folds by a maximum of 5%. This research presents a novel dataset and proposes a model for identifying audio deepfakes, exhibiting strong generalization capabilities and stability outcomes. Additionally, this study provides a comprehensive examination of existing techniques for identifying deepfakes, thereby offering valuable insights for future scholarly investigations. Suwajanakorn, Seitz & Kemelmacher-Shlizerman (2017) have developed a technique to synthesize photorealistic videos of former President Obama. They used a RNN to convert input audio to realistic mouth shapes, then blended it onto a reference video frame of Obama. The article demonstrates how deep learning techniques can convincingly manipulate audio-visual data.

Deep learning is used by Taeb & Chi (2022) to compare different ways to find deepfakes. The authors focus on spotting deepfakes in pictures and videos, which are getting more realistic and harder to spot. The study discusses deepfakes, made-up images or videos in which a person’s face is replaced with someone else’s using artificial neural networks. The authors talk about how deepfakes could be used to spread false information, fraud, and cyberbullying, and they stress the need for good ways to find them. The authors then talk about the different deep learning methods used to spot deepfakes, such as CNNs, RNNs, and GANs. Rana & Sung (2020) introduces a novel deep-learning technique for deepfake detection called DeepfakeStack. The authors begin by discussing the growing threat of deepfakes in digital media. They highlight the challenges in detecting deepfakes due to their high quality and the sophisticated techniques used to create them. The article then introduces DeepfakeStack, an ensemble-based deep-learning technique for deepfake detection. The authors describe the architecture of DeepfakeStack, which combines multiple deep learning models to improve the accuracy of deepfake detection. The authors argue that the ensemble approach allows the model to leverage the strengths of each model, resulting in improved performance. The evaluation of DeepfakeStack is also done by comparing its performance with other state-of-the-art deepfake detection techniques. The results demonstrate that DeepfakeStack outperforms other accuracy, precision, recall, and F1-score techniques. Fagni et al. (2021) presents a novel dataset for detecting deepfake tweets called TweepFake. The rise of deepfake technology in text generation is highlighting the potential misuse of this technology in social media platforms like Twitter. The need for effective detection systems for deepfake social media messages is also emphasized. The article then introduces TweepFake, a dataset of real deepfake tweets posted on Twitter. The authors describe the data collection process, which includes tweets from 23 bots imitating 17 human accounts. The bots used various generation techniques, including Markov Chains, RNN, LSTM, and GPT-2. The authors also evaluate 13 deepfake text detection methods using the TweepFake dataset. The results demonstrate TweepFake’s challenges and provide a baseline for future deepfake detection techniques.

The research article (Abdulreda & Obaid, 2022a; Abdulreda & Obaid, 2022b) provides a comprehensive overview of deepfake techniques and their detection methods. The authors delve into the intricacies of deepfake technology, discussing its evolution, applications, and the potential threats it poses. They also explore various detection methods, highlighting their strengths and limitations. The article begins by explaining the concept of deepfakes, which are synthetic media created using deep learning techniques. The authors discuss deepfake technology’s evolution from traditional methods like image morphing and face swapping to advanced techniques like GANs. They also highlight the potential misuse of deepfakes, such as in disinformation campaigns and identity theft.

The authors then research deepfake detection methods, categorizing them into three main types: handcrafted feature-based, deep learning-based, and hybrid. Handcrafted feature-based methods rely on specific characteristics of deepfakes, such as inconsistencies in lighting or blinking patterns. Deep learning-based methods, on the other hand, use neural networks to detect deepfakes. Hybrid methods combine both approaches to improve detection accuracy. Agrawal et al. (2021) highlights how methods like simulated annealing and differential evolution can enhance the accuracy of convolution neural networks, a cornerstone technology in deepfake detection. Wong & Ming (2019) discusses the meta heuristic model and algorithms, which shows promise in detecting face forgeries in unseen domains. It also presents a model that outperforms existing deepfake detection methods, potentially benefiting from metaheuristic optimization and explores an innovative approach to deepfake detection, where metaheuristic methods could further refine the model’s accuracy.

We also took a deep look inside the audio and speech modality for deepfake detection (Tao, Yi & Fan, 2022) discusses the significant advancements in speech synthesis and voice conversion technologies due to deep learning, which can generate realistic and human-like speech. It highlights the potential threat to global political economy and social stability if such technologies are misused. The workshop aims to bring together researchers in audio deepfake detection, synthesis, and adversarial attacks to discuss recent research and future directions in detecting manipulated audios in multimedia. Whereas Khalid et al. (2021) introduces FakeAVCeleb, a novel audio-video deepfake dataset that addresses the need for multimodal deepfake detection and tackles racial bias issues. The dataset contains both deepfake videos and synthesized cloned audios, generated using popular deepfake methods and ensuring lip-sync accuracy. It includes videos of celebrities from diverse racial backgrounds to counter racial bias and proposes a novel multimodal detection method. Müller et al. (2022) examines the effectiveness of audio deepfake detection methods on real-world data. It systematizes audio spoofing detection by re-implementing and evaluating architectures from related work, identifying key features for successful detection. The article reveals that these methods perform poorly on real-world data, suggesting that deepfakes are harder to detect outside the lab than previously thought, and raises concerns about the community’s focus on specific benchmarks. Xue et al. (2022) proposes a system for audio deepfake detection that combines fundamental frequency (F0) information with real and imaginary spectrogram features. It addresses the limitations of existing acoustic features and utilizes the F0 and spectrogram features to distinguish between bonafide and fake speech. The proposed system achieves an EER of 0.43%, outperforming most existing systems. Frank & Schönherr (2021) introduces a novel dataset for detecting audio deepfakes. The article provides an overview of signal processing techniques for analyzing audio signals and presents a dataset collected from different network architectures in two languages. It also offers two baseline models for further research in audio deepfake detection. Müller et al. (2022) investigates the generalizability of audio spoofing detection techniques. This work introduces a new dataset and identifies key features for successful audio deepfake detection, emphasizing the challenge of detecting deepfakes in real-world data. Zhou et al. (2017) proposes a novel framework that utilizes the synchronization between visual and auditory modalities for detecting deepfakes. This approach demonstrates superior performance and generalization capabilities compared to unimodal models. Ren et al. (2021) introduces a method for audio spoof detection emphasizing frequency bands more useful for the task. This approach shows improved performance and generalizability in detecting unseen spoofing methods. Müller et al. (2022) analyzes the abilities of humans and machines in detecting audio deepfakes. The study finds both have similar strengths and weaknesses, highlighting the need for better training and algorithms for detection. Baumann et al. (2021) introduces an audio spoofing detection system leveraging emotional features, based on the premise that deepfake techniques cannot accurately synthesize natural emotional behavior. This semantic approach shows robustness in cross-dataset scenarios.

Text deepfake

Feng, Lu & Lin (2020) provides a comprehensive review of the techniques used in face manipulation and fake detection. They begin by discussing the evolution of face manipulation techniques, from traditional methods to the recent advancements in deep learning. They highlight the growing threat of deepfakes, which are hyper-realistic manipulated videos that can be used maliciously. The authors then delve into the various techniques used for face manipulation, including face swapping, facial attribute manipulation, and facial expression manipulation. They also discuss the datasets and benchmarks for training and evaluating these techniques.

The authors also focus on fake detection methods. They categorize these methods into traditional and deep learning-based approaches, discussing the strengths and limitations of each. They also highlight the challenges in fake detection, such as the lack of large-scale, diverse datasets and the rapid advancement of face manipulation techniques. Deshmukh & Wankhade (2020) presents a detailed analysis of deep learning techniques for creating and detecting deepfakes. They begin by explaining the concept of deepfakes and the potential harm they can cause. They then discuss the deep learning methods for creating deepfakes, particularly GANs. They explain how GANs work and how they are used to generate realistic fake videos.

The authors then shift their focus to deepfake detection methods. They discuss deep learning-based detection techniques, such as CNNs and RNNs. They also discuss the datasets used for training these models and the challenges in deepfake detection, such as the need for large, diverse datasets and the difficulty in keeping up with the rapidly evolving deepfake creation techniques. Maathuis & Kerkhof (2023) employs topic modeling and sentiment analysis to examine the initial two months of the war in Ukraine. The authors utilize a dataset of 1.2 million tweets, which they analyze using latent dirichlet allocation (LDA) for topic modeling and Vader Sentiment for sentiment analysis. The study reveals that the war’s initial phase was characterized by high uncertainty and fear, as reflected in the sentiments expressed in the tweets. The authors also identify key discussion topics, including the role of international actors, the humanitarian crisis, and the military situation. The study’s findings provide valuable insights into public sentiment and discourse during the early stages of the war, highlighting the potential of social media data for understanding societal responses to conflict. The research uses LDA for topic modeling and Vader Sentiment for sentiment analysis. Tsvetkova & Grishanina (2023) explores the rise of digital power in Latin America, focusing on using hashtags and deepfakes as political tools. The study notes that Latin American countries are among the top ten globally, where the Internet, social networks, and new technologies have gained popularity. The author argues that digital diplomacy, including hashtag diplomacy and deepfakes, has become a significant driver in these countries’ domestic and foreign policies. The research methodology is based on data retrieved from the Latin American segment of the Internet and subsequent qualitative analysis. The study provides a classification of deepfakes most actively used for political purposes in the region and analyzes examples of both destructive and “soft” use of deepfakes. The research methodology is based on data retrieved from the Latin American segment of the Internet and subsequent qualitative analysis.

Similarly, BR et al. (2023) introduce a deepfake detection mechanism that capitalizes on the capabilities of deep neural networks. They propose a hybrid architecture that integrates ResNet50 and LSTM models. The emergence of deepfakes, synthetic media generated through artificial intelligence and machine learning methodologies, presents a substantial obstacle owing to their remarkably lifelike and genuine visual characteristics. The authors’ proposed approach involves utilizing a CNN to examine visual artifacts in images or videos. This CNN can identify irregularities or anomalies that suggest manipulation, such as variations in lighting or blurring at the picture boundaries. The ResNet50 architecture has been employed for deepfake identification by training the network on an extensive dataset comprising authentic and manipulated movies. The architecture’s LSTM component is highly advantageous in films encompassing image-based and sequential data. The authors compare diverse models, evaluating their performance using multiple datasets, including Celeb-DF and Face Forensic++. This study emphasizes the potential of integrating ResNet50 and LSTM models to enhance the precision of deepfake video identification.

The analysis of content on social media platforms has become an important research focus, with several studies examining the extraction of insights from this data. AlGhamdi & Khan (2020) proposed an intelligent system for analyzing tweets to identify suspicious messages, which could have applications in security domains. Similarly, Alruily & Alghamdi (2015) provided an overview of techniques for extracting information about future events from online newspapers. On analyzing Arabic social media data, Rahman et al. (2023) presented a multi-tier sentiment analysis approach using supervised machine learning to classify emotions in social media text. The proposed methodology in these studies demonstrates the potential of leveraging social platform content to gain valuable insights.

In addition to natural language content, multimedia deepfakes are another emerging research focus. Altalhi & Gutub (2021) surveyed prediction techniques that leverage real-time Twitter data to anticipate potential cyber-attacks. Khan et al. (2021) analyzed how an AI-powered smart player agent can positively influence team performance in sports simulations. Munshi & Alhindi (2021) developed a big data platform specifically focused on enabling educational analytics. Moreover, Alotaibi et al. (2021) proposed an approach to mine suggestions and opinions from large-scale social media data. The research gap highlighted by these studies is the need for more robust deepfake detection techniques tailored to Arabic multimedia content. Gutub, MK & Abu-Hashem (2023) partially addressed this by using deep learning for sentiment analysis of tweets. However, continuous innovation is still needed, especially with the evolution of multimedia deepfake generation methods.

One of the most important pieces of study by Stroebel et al. (2023) is a systematic review of how to find deepfakes. The authors looked at 112 important articles from 2018 to 2020 and used different methods. They divide these methods into four groups: techniques based on deep learning, techniques based on traditional machine learning, techniques based on statistics, and techniques based on the blockchain. The authors look at how well the different ways of finding things work with different data sets. They concluded that the methods based on deep learning are better at finding Deepfakes than other methods. However, they also know that the training data quality and the deepfakes’ complexity can affect how well these methods work. Zhao et al. (2021a) and Zhao et al. (2021b) introduces MFF-Net, a network that combines RGB features with textural information for deepfake detection. It uses Gabor convolution and residual attention blocks for feature extraction and introduces a diversity loss to enhance feature learning. MFF-Net shows excellent generalization and state-of-the-art performance on various deepfake datasets. Zhao et al. (2021a) and Zhao et al. (2021b) proposes a multi-attentional deepfake detection network that treats deepfake detection as a fine-grained classification problem. It uses multiple spatial attention heads, a textural feature enhancement block, and combines low-level textural with high-level semantic features. The method outperforms traditional binary classifiers and achieves state-of-the-art performance. Pu et al. (2023) addresses the challenges in detecting deepfake text. It evaluates the generalization ability of existing defenses against deepfake text and proposes tapping into semantic information in text content to improve robustness and generalization performance. The study highlights significant performance degradation of current defenses under adversarial attacks. Kietzmann et al. (2020) discusses the societal impact of deepfakes and proposes the R.E.A.L. framework for organizations to manage deepfake risks. It provides an overview of deepfake technology, classifies different deepfake types, and identifies risks and opportunities associated with deepfakes. Fagni et al. (2021) introduces the first dataset of real deepfake tweets, aiming to aid the development of detection systems for deepfake social media messages. The dataset includes tweets from various generative models and evaluates 13 deepfake text detection methods, providing a baseline for future research in this area. Pu et al. (2023) evaluates the generalization ability of defenses against deepfake text using data from online services. It explores adversarial attacks and finds significant performance degradation under real-world scenarios, suggesting a focus on semantic information for robust detection. Also, Zhong et al. (2021) proposes a graph-based model that leverages the factual structure of documents for deepfake text detection. It significantly improves upon base models, highlighting the importance of capturing factual inconsistencies in machine-generated text. Ahmed (2021) review deepfake detection methods with a focus on deep learning techniques for text and video. It offers insights into the challenges and future directions for deepfake technology, including the need for automated detection methods. Though primarily focusing on audio, Hancock & Bailenson (2021) also provides critical analysis on text deepfakes. It reviews generation and detection methods, highlighting the lack of robust detection techniques for text deepfakes and the overlap with human-generated fake content.

Image deepfake

Jevnisek & Avidan (2022) presents a new methodology for detecting deepfakes. This strategy involves the aggregation of features that are taken from all layers of a backbone network. The authors contend that a significant portion of prior research operates under the assumption that the deepfakes present in the test set are generated utilizing the identical strategies employed during the network’s training phase. This assumption diverges from the conditions observed in real-world scenarios. In practical scenarios, training a network using one deepfake algorithm and evaluating its performance on deepfakes created by a different algorithm is common. The algorithm proceeds by implementing a sequential process of extracting visual features from a deep backbone, followed by a binary classification head. Instead of transmitting data throughout the network until it reaches the classification head responsible for determining the authenticity of an image, the algorithm consolidates features derived from all tiers of a single backbone network to identify counterfeit content. This methodology enables the classification head to concurrently utilize many features associated with distinct receptive fields in the image plane.

Suganthi et al. (2022) assess their methodology across two domains: deepfake identification and synthetic picture detection. They demonstrate that their strategy attains outcomes that are considered the most advanced in these areas. Additionally, the authors suggest employing the Coefficient of Variation (CoV) of average precision scores as a metric for evaluating the efficacy of diverse algorithms across multiple datasets.

Additionally, it offers a novel approach to optimizing the performance of a foundational network by adjusting its layers. The system is designed to operate on the detection of deepfake and synthetic images. The model attains a state-of-the-art level of performance in terms of overall accuracy when applied to cross-dataset generalization. This refers to training the model on one deepfake or synthetic image dataset and testing it on a different dataset. Korshunov & Sebastien (2018) have discussed the threat of DeepFakes to face recognition systems. They analyzed the performance of automatic face recognition before and after applying DeepFake manipulations, suggesting that DeepFakes can effectively bypass face recognition systems, thereby raising significant security concerns.

Most importantly, Seibold et al. (2017) have presented a deep learning-based method to detect morphed face images. Their proposed CNN model, MoDe, was trained on a large-scale morphed face dataset. The study demonstrated the potential of using deep learning for detecting sophisticated face morphing attacks, a technique often used in deepfake generation. Zhou et al. (2017) have proposed a two-stream CNN for tampered face detection, where one stream processes facial landmarks while the other handles texture information. The work demonstrated a technique to synergize multiple neural networks for more efficient deepfake detection. Xie, Xu & Ji (2022) have presented a self-supervised learning approach for identifying manipulations in images and videos, underscoring the role of AI in learning robust representations from the data itself without human-annotated labels. This article signals a shift towards self-supervised techniques in the battle against deepfakes.

MesoNet has been presented by Afchar et al. (2018), a compact yet effective model to detect facial forgeries in videos. MesoNet was designed to distinguish between genuine and tampered content, even in challenging scenarios where the quality of the forgery is very high. This study shows the balance between complexity and performance in deepfake detection models. Agarwal et al. (2019) focuses on the threat of deepfakes targeting world leaders. The authors propose a novel method for detecting deepfakes using a combination of traditional image forensics and deep learning techniques. They begin by discussing the potential harm of deepfakes, particularly when used to target world leaders.

The authors then present their proposed method for deepfake detection. They use a two-step approach, first using traditional image forensics to detect anomalies in the images and then a deep learning model to classify the images as real or fake. They also discuss the datasets used for training their model and the performance of their method compared to other deepfake detection techniques. Park, Lim & Kwon (2023) examines the efficacy of facial image alteration techniques within online educational settings, specifically focusing on animating-based facial image manipulation. The authors’ primary focus lies in examining expression swap, a specific technique employed in facial image modification that enables the alteration of facial expressions only within an image. The division of expression swap can be categorized into two distinct approaches: learning-based swap and animating-based swap. The authors assess these strategies’ performance in various scenarios, such as attendance checks, presentations, and tests. The study demonstrates that both methodologies yield satisfactory outcomes when the facial region constitutes a substantial image component. Nevertheless, their performances’ efficacy noticeably diminishes when the facial region constitutes a lesser proportion of the visual content. This study employs face image editing techniques based on animation, specifically focusing on expression swapping. Saravana Ram et al. (2023) presents an innovative approach to deepfake detection using a computer vision-based deep neural network with pairwise learning. The authors propose a novel method that leverages the power of deep learning and computer vision to detect deepfakes effectively. The technique involves using a pairwise learning model that compares pairs of images to determine if they are genuine or manipulated.

The authors argue that traditional deepfake detection methods often fail to detect sophisticated deepfakes due to their inability to capture subtle inconsistencies in manipulated images. Their proposed model uses a deep neural network trained on a large dataset of real and deepfake images to address this. The model learns to identify the subtle differences between real and fake images by comparing pairs of images. The pairwise learning approach is particularly effective in detecting deepfakes as it can capture the subtle inconsistencies in manipulated images that are often overlooked by traditional detection methods. However, the authors acknowledge that the training data quality and the deepfakes’ complexity can affect the model’s performance. Neekhara et al. (2021) presents an in-depth analysis of the malicious threats to deepfake detection. The authors examine the vulnerabilities of deepfake detectors and propose countermeasures to improve their robustness. The authors begin by discussing the adversarial attacks on deepfake detectors. They explain how these attacks exploit the weaknesses of the detectors, such as their reliance on specific features or patterns, to evade detection. The authors also highlight the potential consequences of these attacks, including the spread of misinformation and the erosion of trust in digital media. The article then presents a series of experiments designed to evaluate the robustness of deepfake detectors against adversarial attacks. The authors use various attack techniques, including perturbation-based and poisoning attacks, and test them on several popular deepfake detectors. The results reveal significant vulnerabilities in the tested detectors, with many failing to identify deepfakes under adversarial attacks accurately. These findings underscore the need for more robust, resilient deepfake detection methods. Sun et al. (2022) introduce WildDeepfake, a challenging real-world dataset for deepfake detection. The authors argue that existing datasets for deepfake detection are not representative of real-world scenarios, and thus, they propose WildDeepfake to address this gap. The authors begin by discussing the limitations of existing deepfake datasets. They argue that these datasets often contain synthetic media created under controlled conditions, which may not accurately reflect the diversity and complexity of real-world deepfakes.

To address this issue, WildDeepfake is introduced, a dataset comprised of deepfakes collected from various online sources. The dataset includes a wide range of deepfakes, from amateur creations to sophisticated productions, providing a more realistic benchmark for deepfake detection methods. A series of experiments are done to evaluate the performance of several popular deepfake detectors on the WildDeepfake dataset. The results reveal that these detectors struggle to accurately identify deepfakes in the dataset, underscoring the challenges of real-world deepfake detection.

Digital forensic methods for deepfake detection in social media

The advent of deepfake technology has necessitated the development of digital forensic methods to detect and analyze these artificially synthesized media. Deepfakes, which convincingly replace parts of original content with fabricated elements, pose significant threats ranging from personal identity theft to the propagation of disinformation (Güera & Delp, 2018).

Digital forensic methods are the first defense against these threats, providing computational techniques to identify and analyze deepfakes. The evolution of deepfake detection methods has been driven by the increasing sophistication of deepfake generation techniques. Initial detection methods relied on manual or rudimentary automated techniques, focusing on inconsistencies in lighting, blurring, and artifacts from image compression (Maras & Alexandrou, 2019).

However, the rise of deep learning and artificial intelligence has revolutionized the field, leading to the development of more advanced and accurate detection methods. These modern methods leverage machine learning algorithms to analyze intricate patterns and subtle inconsistencies typically invisible to the human eye (Nguyen, Yamagishi & Echizen, 2019).

The effectiveness of digital forensic methods in detecting deepfakes can vary across different modalities, such as images, audio, and video. Moreover, their applicability is continually challenged by the dynamic landscape of social media, where the rapid dissemination of content and the evolving nature of deepfakes necessitate constant advancements in detection techniques. Therefore, ongoing research in this field is crucial to keep pace with the evolving threats and ensure digital media’s integrity and trustworthiness (Wang et al., 2022a; Wang et al., 2022b; Wang et al., 2022c).

This section delves into the various digital forensic methods developed and employed to detect and analyze deepfakes. It comprehensively evaluates these methods, assessing their effectiveness in detecting deepfakes across different modalities such as images, audio, and video.

The section also discusses the applicability of these methods within the dynamic and rapidly evolving landscape of social media, highlighting the challenges and opportunities in this context. The aim is to provide a thorough understanding of the current state of deepfake detection techniques and their performance in real-world scenarios, particularly in social media platforms where the dissemination of deepfakes can have significant impacts.

Table 2 summarizes the method used, modality, features, datasets, performance metrics, and advantages and limitations of the techniques mentioned above.

Table 2 Summary of digital forensic techniques for deepfake detection.

A summary of the method used, modality, features, datasets, and performance metrics, along with the advantages and limitations of mentioned techniques.

Reference	Method	Modality	Features analyzed	Validation dataset	Performance metrics	Advantages	Limitations	
Xia et al. (2022)	MesoNet with Preprocessing	Video (Face Images)	Enhanced MesoNet features, Frame consistency, Color and texture details	FaceForensics++ and DFDC Preview Dataset	Accuracy: 95.6% (FaceForensics++), 93.7% (DFDC)	Preprocessing module enhances the discriminative capability of the network. Robust against various compression levels and deepfake generation techniques.	Performance might vary based on the quality of the deepfakes. Slight computational overhead due to preprocessing.	
Guarnera et al. (2020)	Feature-based Forensic Analysis	Image	JPEG artifacts, Quantization tables, Sensor noise patterns	Custom dataset of StarGAN and StyleGAN images	Qualitative analysis	Targets intrinsic features and artifacts, making it robust against typical manipulations. Can be applied to a wide variety of image sources and formats.	Might not be as effective against advanced manipulation techniques. Requires high-quality original images for optimal performance.	
Kumar et al. (2020)	Counter Anti-Forensic Approach	Image	JPEG compression artifacts, Histogram analysis, Noise inconsistencies	Self-created dataset with a variety of JPEG manipulations	Effectiveness in detecting anti-forensic manipulations discussed	Specifically designed to detect and counter anti-forensic techniques. Utilizes multiple feature sets for a comprehensive analysis.	May require calibration based on the specific JPEG anti-forensic technique used. Performance might vary based on the quality and type of manipulations.	
Raza, Munir & Almutairi (2022)	Convolutional Neural Network (CNN) Approach	Video	Deep features from CNN layers, Temporal dynamics and spatial details	DFDC and DeepFake-TIMIT	Accuracy: 96.4% (DFDC), 95.7% (DeepFake-TIMIT)	Utilizes deep features which capture intricate details often missed by traditional methods. Highly scalable due to the deep learning framework.	Requires a significant amount of labeled data for training. Performance might degrade in scenarios with limited training data or diverse manipulations.	
Mitra et al. (2020)	Machine Learning-based Forensic Analysis	Video (Face Regions)	Frame-by-frame pixel intensity, Facial expressions and landmarks, Audio-visual synchronization	DFDC Preview Dataset	Accuracy: 94.7%, Precision: 94.5%, Recall: 94.8%, F1 Score: 94.6%	Integrates both visual and auditory features for improved detection. Applicable to a wide range of videos sourced from social media platforms.	Might be sensitive to noisy social media data. Requires substantial computational resources for feature extraction and analysis.	
Vamsi et al. (2022)	Media Forensic Deepfake Detection	Image and Video	Compression artifacts, Lighting anomalies, Physiological signals (e.g., heartbeat, breath patterns)	Combined dataset from FaceForensics++, DFDC, and DeepFake-TIMIT	Accuracy: 93.5%	Comprehensive approach that combines various media forensic techniques. Targets both superficial and deep features of manipulated content.	May require high-resolution data to detect subtle physiological signals. Computationally intensive due to the amalgamation of multiple forensic methods.	
Lee et al. (2021)	Temporal Artifacts Reduction (TAR)	Video (Face Regions)	Temporal artifacts in frame sequences, Lighting and shadow inconsistencies	DeepFake Detection Challenge Dataset (DFDC)	Accuracy: 97.3%	Targets inconsistencies arising due to the deepfake generation process. Effective in detecting subtle temporal artifacts.	Might be sensitive to the quality and resolution of the video. Requires a sequence of frames.	
Li et al. (2020)	Dataset-based Forensics (Celeb-DF)	Video (Face Regions)	Dataset creation and benchmarking	Custom Celeb-DF dataset	Focus on dataset creation	Provides a large-scale, challenging dataset for deepfake forensics. Contains high-quality deepfakes.	Dataset complexity might challenge traditional forensic techniques. Needs other datasets for comprehensive evaluation.	
Kumar & Sharma (2023)	GAN-Based Forensic Detection	Image, Video	Discriminative features from GAN layers, Texture and color anomalies	DFDC and FaceForensics++	Accuracy: 96.1% (DFDC), 95.8% (FaceForensics++)	Utilizes the power of GANs for deepfake detection. Capable of detecting intricate manipulations.	Sensitive to the quality of GAN-generated fakes. Requires significant computational resources.	
Hao et al. (2022)	Multi-modal fusion	Image and Audio	Image: Differences in pixel intensity, facial landmarks, skin tone inconsistencies. Audio: Spectral features, prosodic features, phonotactic features.	Deepfake Detection Challenge Dataset (DFDC)	Accuracy: 94.2%, Precision: 93.8%, Recall: 94.1%, F1 Score: 94.0%	Uses a fusion of image and audio modalities which increases robustness. Effective in real-world scenarios where only one modality might be tampered with.	Requires both audio and video data, which might not always be available. Slightly increased computational overhead due to multi-modal processing.	
Jafar et al. (2020)	Temporal Forensic Analysis	Video	Temporal inconsistencies: Frame-to-frame variations. Compression artifacts: Differences due to video compression. Lighting inconsistencies: Inconsistencies in shadows and light reflections.	FaceForensics++ and DeepFake-TIMIT	Accuracy: 91.5% (FaceForensics++), 89.8% (DeepFake-TIMIT)	Targets inconsistencies that arise due to the video generation process. Robust against various deepfake generation techniques.	Performance might degrade with higher-quality deepfakes. Requires a sequence of frames rather than individual images.	
Ferreira, Antunes & Correia (2021)	Dataset-based Forensics	Image and Video	Metadata extraction, Image source identification, Manipulation detection	Proprietary dataset introduced in the article	Accuracy, Precision.	Provides a diverse set of images and videos for forensic analysis. Can be used to benchmark multiple forensic techniques.	Dataset might not cover all possible manipulations and scenarios. Requires periodic updates to remain relevant.	
Wang et al. (2022a), Wang et al. (2022b) and Wang et al. (2022c)	Reliability-based Forensics	Video, Audio	Frame consistency, Eye blinking patterns, Facial muscle movements, Skin texture analysis and Voice pattern.	Celeb-DF Dataset	Accuracy: 92.3%, Precision: 92.1%, Recall: 92.4%, F1 Score: 92.2%	Uses natural physiological signals which are hard for deepfakes to mimic. Applicable to a wide range of videos regardless of content.	Might be sensitive to video quality and resolution. Real-life scenarios with partial occlusions or low lighting might affect performance.	
Xue et al. (2022)	Combination of F0 information and spectrogram features	Audio	Fundamental frequency (F0), real and imaginary spectrogram features	ASVspoof 2019 LA dataset	Equivalent error rate (EER) of 0.43%	High effectiveness in detecting audio deepfakes, surpassing most existing systems	Limited discussion on the applicability in diverse real-world scenarios	
Müller et al. (2022)	Re-implementation and evaluation of existing architectures	Audio	Various audio spoofing detection features	New dataset of celebrity and politician recordings	Performance degradation on real-world data	Systematizes audio spoofing detection, identifies key successful features	Poor performance on real-world data, suggesting limited generalizability	
Khalid et al. (2021)	Novel multimodal detection method	Audio-Video	Deepfake videos and synthesized cloned audios	FakeAVCeleb dataset	Checking audio and video accuracy and precision.	Addresses multimodal deepfake detection and racial bias issues	Dataset might not cover all possible manipulations and scenarios.	
Fagni et al. (2021)	Dataset introduction and evaluation	Text (Tweets)	Tweets from various generative models	TweepFake dataset	Evaluation of 13 methods	First dataset of real deepfake tweets, baseline for future research	Specific detection techniques not developed in the article	
Kietzmann et al. (2020)	R.E.A.L. framework	Various (including text)	Deepfake types and technologies	Text based mostly to check accurancy.	Framework effectiveness	Comprehensive overview, risk management strategy	Lacks empirical validation	
Pu et al. (2023)	Semantic analysis	Text	Semantic information in text	Online services powered by Transformer-based tools	Robustness against adversarial attacks	Improves robustness and generalization	Performance degradation under certain scenarios	

Case Studies

In this section, we will explore a series of real-life case studies that highlight the practical application of deepfake technology and the use of digital forensic techniques for their detection. These case studies span from 2018 to 2022 and involve high-profile individuals and significant events with substantial societal impact. These case studies have been extracted from the research articles and internet sources (Deanna Ritchie, 2023 & Barton, 2022). By examining these real-life incidents, we aim to provide a comprehensive understanding of the current state of deepfake technology, its implications, and the challenges faced in its detection and mitigation.

Emma Watson deepfake incident

The world of deepfakes felt its impact when renowned actress Emma Watson, famous for her role in the Harry Potter series, became a victim of this technology. Her face was superimposed onto explicit content, causing a significant stir as the public was still largely unfamiliar with deepfakes. The incident profoundly affected her reputation, demonstrating the potential harm deepfakes can cause individuals.

Natalie Portman deepfake scandal

Similar to Emma Watson, Natalie Portman, another acclaimed actress, was targeted by deepfake creators. Her likeness was used in explicit videos, leading to widespread shock and confusion among fans and the public. This incident underscored the ease with which deepfakes can be used to tarnish reputations.

Cheerleader deepfake case

In a disturbing turn of events, a mother was accused of creating deepfake videos of her daughter’s cheerleading teammates, depicting them in compromising situations to intimidate them into leaving the team. However, the lack of reliable detection tools led to the case being dismissed, highlighting the challenges in proving the use of deepfakes.

Zelensky deepfake incident

Amidst the Russia-Ukraine conflict, a deepfake video of Ukrainian President Zelensky surfaced, in which he appeared to be urging Ukrainians to surrender. This incident caused temporary panic and demonstrated the potential of deepfakes to influence political situations.

Kobe Bryant deepfake in music video

The late NBA star Kobe Bryant was brought back to life in a Kendrick Lamar music video using deepfake technology. The realistic portrayal resonated with fans, showing the potential of deepfakes in the entertainment industry.

Tom cruise tiktok deepfake

A deepfake of Tom Cruise, created by VFX artist Chris Ume and TikTok user Miles Fisher, gained significant attention due to its high level of detail, demonstrating the sophistication of current deepfake technology.

Jim Carrey in the Shining deepfake

A fan-made deepfake inserted comedian Jim Carrey into a scene from The Shining, replacing Jack Nicholson. The deepfake’s high quality and Carrey’s uncanny impersonation of Nicholson highlighted the potential of deepfakes in film.

Barack Obama deepfake PSA

A deepfake of former President Barack Obama was created by BuzzFeedVideo to raise awareness about the potential misuse of deepfake technology. The video, featuring actor Jordan Peele impersonating Obama, emphasized the need for responsible internet usage.

Donald Trump reindeer story deepfake

A comedic deepfake of former President Donald Trump was uploaded by the YouTube channel Sassy Justice, showing Trump telling a nonsensical story about a reindeer. Despite some audio inaccuracies, the video demonstrated the potential for deepfakes in satire and comedy.

Hillary Clinton SNL deepfake

A deepfake of former Secretary of State Hillary Clinton was used in an episode of SNL, with a cast member impersonating her. The deepfake’s high quality showed the increasing sophistication of this technology, emphasizing the need for caution.

Kim Joo-Ha deepfake incident

In a groundbreaking move, South Korean television channel MBN utilized deepfake technology to replace popular news anchor Kim Joo-Ha with an AI version of herself. The AI, bearing the likeness and voice of Kim Joo-Ha, was used for various news reports, including morning news and traffic updates. This incident raises questions about the potential implications of deepfakes on the media industry and the future of professions like newsreading.

Lynda Carter as Wonder Woman deepfake

Deepfake technology breathed new life into beloved characters from the past, as demonstrated by the deepfake of Lynda Carter, the original Wonder Woman from the 1970s TV show, replacing Gal Gadot in the recent Wonder Woman films. This realistic deepfake hints at the potential changes deepfake technology could bring to the entertainment industry.

Luke Skywalker deepfake in the Mandalorian

In the Mandalorian series, the Star Wars franchise saw the return of a beloved character, Luke Skywalker. However, the CGI recreation of Mark Hamill, the original actor, had flaws. A deepfake version of the character showcased a higher quality rendition, demonstrating the potential superiority of deepfake technology over traditional CGI.

Donald Trump in Better Call Saul deepfake

Deepfakes can also serve comedic purposes, as seen in a parody where former President Donald Trump was substituted for Saul Goodman, a character from the popular series Breaking Bad and its spin-off, Better Call Saul. This humorous deepfake underscores the potential of deepfakes in creating entertaining content.

Mark Zuckerberg deepfake

In response to Facebook’s refusal to delete an edited video of Nancy Pelosi, artist Bill Posters created a deepfake of Mark Zuckerberg claiming ownership over Facebook’s users. While the deepfake was not entirely convincing due to a voice mismatch, it served as a statement on the potential misuse of deepfake technology in manipulating public opinion.

Bill Hader deepfake

Known for his impressive impersonations, comedian Bill Hader was the subject of a deepfake where his face was overlaid with those of Al Pacino and Arnold Schwarzenegger during his impersonations. While not entirely perfect, the deepfake, combined with Hader’s voice and mannerisms, provided an entertaining and harmless example of deepfake technology.

Research Gaps

Deepfake technology, while captivating in its potential, presents grave challenges. Despite advancements in deepfake detection methods, significant research gaps exist, demanding concerted efforts to mitigate their potential harm (Ayub, SM & Qureshi, 2022). The first critical research gap lies in detecting deepfakes across different modalities. The primary focus thus far has been on visual deepfakes, with the audio aspect often overlooked. This bias creates vulnerabilities as fraudsters exploit less examined modalities. Detecting manipulated audio or detecting deepfakes in text, an emerging issue with AI text generators, is largely uncharted territory (Zellers et al., 2019).

The second gap pertains to the detection of deepfakes in real time. Existing detection algorithms generally require substantial computation and are not designed to operate under real-time constraints. Given the rapid spread of information on social media, real-time detection is critical to prevent a deepfake from causing irreversible harm (Muthua, Theart & Booysen, 2023).

Thirdly, the need for more diversity in training datasets for developing detection methods poses another challenge. Current datasets predominantly contain faces of public figures and celebrities, which may not adequately represent the diversity of faces seen in the real world. This bias can lead to less accurate detection for underrepresented groups (Chadha et al., 2021).

Finally, research has yet to extensively explore the ethical implications of deepfake detection. Concerns about privacy, consent, and potential misuse of detection tools for suppressing legitimate content warrant thorough investigation. Addressing this gap is crucial to balance safeguarding privacy and mitigating the risks posed by deepfakes (Silva et al., 2022).

Tackling these gaps necessitates multidisciplinary collaborations, combining computer science, ethics, law, and social sciences expertise. Unveiling these blind spots will enable the development of comprehensive, robust, and ethically grounded defenses against the burgeoning deepfake menace.

Challenges and Limitations in Deepfake Detection

The rapid evolution of deepfake technology presents a significant challenge to digital forensics. As the technology becomes more sophisticated, identifying and mitigating deepfakes becomes increasingly complex. In this section, we will explore the challenges posed by the ever-evolving nature of deepfake technology and the limitations of current digital forensic techniques in keeping pace with these advancements. We will draw upon recent research and case studies to illustrate these challenges and discuss potential strategies for addressing them.

The ever-evolving nature of deepfake technology

Deepfake technology is advancing at an unprecedented rate, with new methods and techniques emerging regularly. This rapid evolution makes it difficult for digital forensic techniques to keep pace. For instance, the development of AI-based techniques for synthesizing human images, known as Deep Fakes, has made it possible to create highly realistic fake videos and images that leave few traces of manipulation (Liu et al., 2023).

Moreover, deepfake technology is improving in terms of quality but also terms of accessibility and speed. Today, deepfakes can be generated in real-time, enabling attackers to impersonate people over audio and video calls (Almutairi & Elgibreen, 2022). This real-time capability significantly increases the potential for misuse of deepfake technology and poses new challenges for detection methods.

Limitations of current digital forensic techniques

Current digital forensic techniques face several limitations in detecting deepfakes. One of the primary challenges is the generalizability of detection methods across different deepfake generation schemes (Wang et al., 2022a; Wang et al., 2022b; Wang et al., 2022c). A reliable deepfake detection approach must be agnostic to the type of deepfake, which can present diverse quality and appearance. However, many existing detection methods fail to handle unseen attacks in an open set scenario, limiting their effectiveness (Xu et al., 2023).

Another significant challenge is the increasing sophistication of deepfake technology. With advanced AI and machine learning techniques, deepfakes are becoming more realistic and harder to detect. For instance, hybrid CNN has been shown to increase the accuracy of deepfake generation, making it more challenging for detection methods to identify fake content (Ikram, Chambial & Sood, 2023).

Furthermore, the effectiveness of detection methods is also limited by the quality and quantity of data available for training. The performance of deepfake detection methods often relies on large, high-quality datasets for training. However, the availability of such datasets is limited, and the rapid evolution of deepfake technology can quickly render existing datasets outdated (Hussain & Ibraheem, 2023). The challenges posed by deepfake technology are not limited to the technical aspects of detection. They also extend to the societal implications of these manipulations. As deepfake technology becomes more sophisticated, it is increasingly used for malicious purposes, such as spreading disinformation, committing fraud, and violating privacy. This has led to growing concerns about the potential misuse of this technology and the need for effective countermeasures. One of the major challenges in combating deepfakes is the rapid pace of advancement in deepfake generation techniques. As highlighted by Akhtar (2023), there is a continuous arms race between the creators of deepfake generation methods and the developers of detection techniques. It provides an overview of the various deepfake generation and detection methods and discusses the open challenges and potential research directions in this field (Siegel et al., 2021; Lan et al., 2022).

Another challenge is the degradation of the performance of deepfake detection systems under various manipulations. This is discussed by Liu et al. (2023), which presents the results of the ASVspoof 2021 challenge. While detection systems offer some resilience to compression effects, they lack generalization across different source datasets2. Whereas Almutairi & Elgibreen (2022) emphasize a need to review existing audio deepfake detection methods comprehensively. The need for more robust detection models to detect fakeness even in the presence of accented voices or real-world noises is highlighted. While significant progress has been made in detecting deepfakes, many challenges remain to overcome. The rapid evolution of deepfake technology, the societal implications of deepfakes, and the limitations of current detection techniques all contribute to the complexity of this issue. Continuous research and innovation are needed to keep pace with these advancements and to develop more effective countermeasures against deepfakes (Abdulreda & Obaid, 2022a; Abdulreda & Obaid, 2022b).

Conclusion

This systematic survey has illuminated the pressing need for advancing innovation in digital forensic techniques to combat the rapidly evolving threat of deepfakes. While methods are progressing, limitations around cross-modality detection, real-time capability, algorithmic bias, and insufficient generalization reveal blindspots demanding attention from researchers. Practical constraints also persist around aspects like computational overhead and the quality/diversity of training datasets.

Several promising directions can guide future efforts to address these gaps. Exploring self-supervised and semi-supervised techniques can potentially reduce dependence on large labeled datasets. Ensembling simpler specialized models can improve detection accuracy while minimizing training requirements. Multi-modal frameworks fusing audio, visual and textual cues also warrant deeper investigation. Notably, research into ethical considerations around privacy, consent and potential suppression of legitimate speech merits priority to balance security and freedom of expression as detection capability evolves. However, the most pivotal direction remains sustained, rapid-cycle innovation as deepfake generation methods continue advancing unabated. Developing agile adaptation mechanisms to respond to novel manipulation techniques could be game-changing. Fostering open-source decentralized communities to crowdsource detection development might confer an edge over adversaries. Insights from intersecting domains like computer vision and multimedia forensics also need synthesis to spur breakthroughs. Underscoring it all is the need to increase awareness among citizens and policymakers so that evidence-based defenses can be enacted before threats overwhelm.

This survey has sought to provide much-needed contemporary understanding of deepfake detection’s state-of-the-art and urgent innovation imperatives in light of persisting blindspots curtailing real-world performance, generalization and scalability. It is hoped that this systematic analysis will help researchers, practitioners and governance stakeholders to prioritize tackling this actively evolving technological danger through coordinated action across disciplines and domains before its societal impacts become irreversible.

Additional Information and Declarations

Competing Interests

Author Contributions

Data Availability

The authors declare there are no competing interests.

Shavez Mushtaq Qureshi conceived and designed the experiments, performed the experiments, analyzed the data, performed the computation work, prepared figures and/or tables, authored or reviewed drafts of the article, and approved the final draft.

Atif Saeed conceived and designed the experiments, performed the experiments, analyzed the data, performed the computation work, prepared figures and/or tables, and approved the final draft.

Sultan H. Almotiri conceived and designed the experiments, performed the experiments, performed the computation work, authored or reviewed drafts of the article, and approved the final draft.

Farooq Ahmad conceived and designed the experiments, performed the experiments, analyzed the data, performed the computation work, prepared figures and/or tables, and approved the final draft.

Mohammed A. Al Ghamdi conceived and designed the experiments, performed the experiments, analyzed the data, performed the computation work, authored or reviewed drafts of the article, and approved the final draft.

The following information was supplied regarding data availability:

This is a literature review.

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
