# Peer review of "Deepfake forensics: a survey of digital forensic methods for multimodal deepfake identification on social media"

_PeerJ Computer Science, doi:10.7717/peerj-cs.2037_

## Round 0.1 · original submission · Major Revisions

The topic addressed by the paper is timely and interesting; however, the paper need major revisions to address the issues raised by the reviewers, the most import of which are as follows:
- although the paper focus on multimodality, it mainly addresses image and video information; inclusion of text and audio should be more extensively addressed;
- bibliography should be integrated;
- the cited papers should be compared based on pre-defined properties/criteria;
- presentation needs to be improved from a linguistic point of view.

Reviewer 2 has suggested that you cite specific references. You are welcome to add it/them if you believe they are relevant. However, you are not required to include these citations, and if you do not include them, this will not influence my decision.

**Language Note:** The Academic Editor has identified that the English language must be improved. PeerJ can provide language editing services - please contact us at [email protected] for pricing (be sure to provide your manuscript number and title). Alternatively, you should make your own arrangements to improve the language quality and provide details in your response letter. – PeerJ Staff

Reviewer 1 ·

Basic reporting

The presented review is actually on a topic that is very trending and has recently been the target of many researchers. As far as I am concerned, the exact topic of multi-modal deepfake methods is not recently reviewed; however there are several serious flaws in the presented article.

Commenting on the structure of the article, as far as I can see, the article consists of the following sections: Abstract, Introduction, Survey Methodology, Research Gaps, and Conclusion. The Survey Methodology section consists of 18 pages covering datasets, technology and forensic methods in social media). I think this is a very bad structure, hard to follow and unfriendly.

In the pdf there are two tables (Table1 and Table2) summarizing the literature yet they were not mentioned within the text. I think the tables would have been a good replacement to a big part of the text as it is easier to see the actual difference between research.

Figures 1, 2, 3 and 4 are very trivial and hardly an addition to knowledge

Experimental design

The title clearly stated that this is a review of multimodal methods, yet throughout the text, the focus on other modalities was minimal.

For instance, text as a modality was suddenly mentioned within the previous work in lines 429 to 438, 507 to 515, 516 to 526 and 554 to 563. There was not proper mention in the introduction.

Yet audio was hardly mentioned. The article needed to have a bigger focus on multi modal domains. The whole text was oriented towards video and images, yet hardly any focus on the other modalities. Only two research in table 2 addressed the audio and video modality again with no mention within the text itself

The datasets need to be placed in a table with a comparison among them and including all modalities as promised in the title. Where are the datasets involving multi modal data?

Validity of the findings

How can you validate Figure 5? improvement in the accuracy of detection over the years? On which dataset? on which modality? This is a very vague chart that is not well supported by text, references or argument.

Additional comments

Reading the whole text without the title would give the impression that this is an image or video deepfake survey with some reference to text within. Yet what was promised within the title and abstract of being a multi-modal deepfake survey was not well delivered within the text.

Reviewer 2 ·

Basic reporting

Dear Authors,
although proposed manuscript has merits, there are some issues that need to be addressed.

My observations are below.

- Abstract should be completely reformulated to highlight main ideas and contributions of the proposed research. Abstract should emphasize a problem that is being solved, importance of the problem, employed methods and achieved results along with methods used in comparative analysis.

- Introduction should be clearly presented to highlight main ideas and motivation behind the proposed research. Please include and clearly state research question and contributions of proposed study in Introduction. Also, please clearly explain what is "beyond state-of-the-art" in the proposed study.

Experimental design

Survey methodology is satisfactory, however I am not sure why authors haven't mentioned SCOPUS database. There are definitely some papers in Scopus which are not included in mentioned databases in the paper.

Additionally, there are many deep fake papers that include metaheuristics and I think that these ones should be particularly mentioned.

Also, some relatively recent papers are missing, e.g. https://peerj.com/articles/cs-881/#fig-6

Validity of the findings

I suggest authors to visually compare papers by using multiple criteria. In this way, manuscript would be more interesting for the readers. Also, if possible statistical tests should be conducted.

Additional comments

Conclusion should be extended to include more details regarding the future work and limitations of proposed study. Limitations should be distinguished between theoretical and practical.

Some references are missing parts, such as pp., publisher, year, etc.

There are some English language and technical errors, please revise, e.g. tables must be aligned on the center of the page, all symbols in equations must be defined, etc.

---

## Round 0.2 · Major Revisions

The revised version of the survey has improved and partially addressed the -reviewers' comments. However, there are still some revisions that should be implemented. The most important one is that still in the survey there is no balance between the different media that the survey claim to consider in the Introduction. Both in the presentation of the datasets, and in the discussion of the methods, a clearer distinction based on the considered modalities should be done.

Reviewer 1 ·

Basic reporting

I think the introduction is very repetitive. The article is still very hard to follow. It needs a section per modality and then a section combining them.

Experimental design

I still think that the survey lacks a balance between the different modalities. I can see that the text and audio have been slightly added, but they do not balance with the focus of the image and video. For instance, the dataset section presents only images and videos, although it was mentioned within the text that there is indeed text research in deepfake. Furthermore, the title promises multi-modality, which emphasizes the use of several modalities in the same research. I don't think this is the focus of the presented survey.

Validity of the findings

no comment

Reviewer 2 ·

Basic reporting

Authors have addressed all raised issues.

Experimental design

Authors have addressed all raised issues.

Validity of the findings

Authors have addressed all raised issues.

Additional comments

Authors have addressed all raised issues.

---

## Round 0.3 · Minor Revisions

In their revision the authors have addressed several issues raised by the reviewers. It is recommended to better outline the main aspects related to multimodality that are considered in the survey, at least in the Introduction.

---

## Round 0.4 · accepted · Accept

The authors have better outlined the aspects related to multimodality considered in the survey.